# Research on Friction and Wear Properties of Rubber Composites by Adding Glass Fiber during Mixing

**DOI:** 10.3390/polym14142849

**Published:** 2022-07-13

**Authors:** Deshang Han, Yihui Chen, Yi Pan, Chuansheng Wang, Dewei Zhang

**Affiliations:** 1College of Electromechanical Engineering, Qingdao University of Science and Technology, Qingdao 266061, China; handeshang@163.com (D.H.); qustchenyihui@163.com (Y.C.); 17806168130@163.com (Y.P.); wangcs202101@163.com (C.W.); 2Shandong Provincial Key Laboratory of Polymer Material Advanced Manufactorings Technology, Qingdao University of Science and Technology, Qingdao 266061, China

**Keywords:** GF/rubber composite, end metal, wear amount, wear ratio

## Abstract

GF/rubber composites have sound insulation characteristics, heat resistance, good corrosion resistance, and high mechanical strength. The compounding machine’s long working hours will inevitably wear the metal on the end face of the compounding machine. The wear of the end face metal will increase the gap between the chamber and the end face, which will lead to material leakage, reduce the mixing effect, and eventually affect the performance of GF/rubber composites. To ensure the implementation of GF/rubber composites, it is necessary to study the frictional wear behavior of GF/rubber composites on metals. In this paper, the effect of blending rubber with different amounts of GF on the frictional wear of metal on the end face was analyzed from the perspective of the formulation process, and the ratio of corrosion wear and abrasive wear was calculated for the first time.

## 1. Introduction

Glass fiber is a kind of material with excellent performance, which has many types, low price, and high mechanical strength. It is widely used in the reinforcement of polymer materials. The composites obtained by adding glass fibers have excellent properties. It has been reported that the composite material of glass fiber and rubber has good comprehensive properties and can improve its anti-skid performance [1,2,3,4,5]. As it is widely used in many fields, glass fiber is increasingly valued. Glass fibers are commonly used as reinforcement materials in composite materials, electrical insulation materials, thermal insulation materials, circuit substrates, etc. An overview diagram of GF is shown in Figure 1.

Li Hong [6] reviewed the development of continuous glass fibers in recent years, focusing on high-performance glass fibers. In particular, the article presents recent advances in the characterization of glass structures using NMR spectroscopy, Raman spectroscopy, and molecular dynamics simulation techniques.

Yang Chuanqi [7] used the mechanical blending method to prepare glass fiber (GF)/natural rubber (NR)/butadiene rubber (BR) composites and studied the influence of glass fiber partially replacing silica on the vulcanization properties, mechanical properties, wear resistance, and skid resistance of composites. The performance of the filler was compared with that of silica and nanometer attapulgite. The results show that adding glass fiber can shorten the curing time and increase the compound’s hardness, tear strength, and wet friction coefficient. When the amount of glass fiber is 6 phr, the comprehensive performance of the binder is higher than that of silica and nanometer attapulgite filler.

Wang Wenjie [8] used glass fiber (GF) as a filler and nitrile butadiene rubber (NBR) as a matrix to prepare NBR/GF composites by the mechanical blending method and studied the effect of the amount of glass fiber on the properties of the composites. It was found that with the increase of the glass fiber dosage, the vulcanization rate of the compound decreased gradually, and the tensile strength and 100% fixed elongation stress of the vulcanized compound increased first and then decreased. The maximum tensile strength and 100% fixed elongation stress were increased by 33% and 550%, respectively.

Chen Jiabin [9] prepared glass fiber/silicon rubber composite materials, studied the failure characteristics of glass fiber/silicon rubber composite materials by the tensile test, analyzed the stress/strain curve of the composite material, and used the finite element method to numerically simulate the composite material. The study found that the glass fiber/silicone rubber composite specimen mainly manifested as a fiber debonding fracture with fiber pulling out under the action of tensile load.

Lv Fuling [10] prepared nano-magnesium hydroxide/glass fiber/silicon rubber composites by mechanical blending. The dispersion state of nano-fillers in the silicone rubber matrix was investigated, and the changing laws of physical and mechanical properties, heat resistance, and flame-retardant properties of composites with different filler dosages were compared. The results show that additional magnesium hydroxide and glass fiber dosage ratios can improve silicone rubber’s physical and mechanical properties. Among them, tensile strength, tear strength, compression set, and hardness have been enhanced. Combining the two fillers improves the composite’s thermal stability and flame retardant properties to a certain extent.

Hao Qiangqiang [11] prepared glass fiber/nitrile rubber composites by the mechanical blending method, studied the effect of glass fiber content and length on the wear resistance and mechanical properties of the composites, and observed the wear surface of the composites with a scanning electron microscope. The results show that the optimum addition mass of chopped glass fibers in the nitrile rubber matrix is 3 phr, and the length is 6 mm. Under these conditions, the prepared composites have the best wear resistance and comprehensive mechanical properties, and the wear surface is the flattest and is smooth.

The preparation of GF/rubber composites is carried out in the compounding machine. With the development of GF/rubber composites, the use time of the compounding machine, which is the equipment for preparing GF/rubber composites, has increased dramatically, bringing about the wear and tear of the compounding machine. This paper studies the frictional behavior of GF/rubber composites with different ratios and end metals during the mixing process.

## 2. Experiment

### 2.1. Instruments

The experimental equipment used in this experiment is shown in Table 1.

### 2.2. Formulation

Table 2 shows the formulae. To calculate the silylation reaction index, adding the C1 group without TESPT as a control is necessary.

The following were used: Butadiene Rubber 9000 (BR9000), China Hainan Natural Rubber Industry Group Co., Ltd., (Haikou, China); Polymeric Styrene Butadiene Rubber (RC2557S), Dongguan Futai Rubber Trading Co., Ltd., (Dongguan, China); Natural Rubber (TSR20), China Hainan Natural Rubber Industry Group Co., Ltd., (Haikou, China); Bis-[γ-(triethoxysilyl)propyl]tetrasulfide (TESPT), Antioxidant 4020, Zinc Oxide ZnO, Stearic Acid SAD, 1,3-Diphenylguanidine (DPG), accelerator CZ, and sulfur S, products of China Henan Longji Chemical Co., Ltd., (Puyang, Chian); Glass Fiber (GF), Wanlong Composite Materials Co., Ltd., (Qingdao, Chian); Silica115MP has a specific surface area of 115 m^2^/g, 7–40 nm particle size, and solid powder, Solvay Silica Co., Ltd., (Brussels, Belgium); GF diameter is 12–23 μm, length is 0.2–0.6 mm, and the surface is not treated.

### 2.3. Mixing Procedure

The GF/rubber composites were obtained after 5 min by mixing according to the mechanical compounding process in Table 3. The obtained GF/rubber composite was over-rolled in the BL-6157 double-roller mill, and the setting was set to a width of the double-roller of 8 mm, and the GF/rubber composite specimens with a thickness of 8 mm were obtained. The obtained GF/rubber composite specimens were cut according to the die of CSM to obtain friction specimens with a diameter of 100 mm and a thickness of 8 mm.

### 2.4. Test Methods

#### 2.4.1. Rubber Processing Properties

This test was performed on the RPA 2000, and the test conditions were set as follows. The silica Durocher Stabilized Cool Down scan test conditions were 0.01 Hz scan frequency, 0.28–40% scan strain range, and 60 °C temperature, and the dynamic modulus *G’* curve with strain was obtained [12,13,14,15,16,17].

(1) Payne effect: The phenomenon in which the elastic modulus of a filler compound decreases as the strain amplitude decreases [18,19,20].

(2) Silanization reaction index: Characterize the extent to which the silanization reaction proceeds. The calculation method is shown in Table 4. The test principle of the degree of silanization reaction is shown in Figure 2.
$$X=\frac{\mathrm{Area}\;\mathrm{of}\;\mathrm{silylation}\;\mathrm{zoon}}{\mathrm{Area}\;\mathrm{of}\;\mathrm{the}\;\mathrm{largest}\;\mathrm{silylation}\;\mathrm{region}}=\frac{\int {G^{\prime }}_{REF}(05)-\int {G^{\prime }}_{S}(05)}{\int {G^{\prime }}_{REF}(05)-\int {G^{\prime }}_{S}(06)}$$

#### 2.4.2. Friction Test

The Hake mixer is shown in Figure 3. The CSM friction and wear testing machine was used to conduct friction experiments.

The CSM was parameterized before the experiment. The rotation speed of the CSM was the same as that of the Hake mixer, which was set to 70 r/min, the pressure was set to 5 N, and the friction time of the CSM was set to 30 min. Previous studies have shown that the most severe wear occurs at the end of the experiment, so the CSM temperature was set to 150 °C. The CSM friction and wear tester is shown in Figure 4 [21,22,23,24,25].

(3) Metal surface observation: The metal grinding head was placed on the stage and the LEXT OLS5000 3D microscope was used to emit a laser to scan the metal surface to obtain metal surface data [26,27,28].

(4) Dispersion: The rubber sample was tested with a DisperGRADER disperser, and the dispersion value was obtained through equipment analysis.

## 3. Mechanistic Study

### 3.1. Silylation Reaction Mechanism

The silanization reaction is divided into two steps, as shown in Figure 5 and Figure 6.

Figure 5 and Figure 6 show the silanization process. Due to limited technical conditions, the quality of ethanol and water produced during blending cannot be measured. To ensure the reliability of the test results, friction and wear experiments were carried out on CSM. According to the proportion of the silanization reaction, the high-temperature ethanol–water mixed vapor was sprayed onto the metal surface. The mixing conditions of the mixer were qualitatively simulated [29,30,31,32,33,34,35,36,37].

### 3.2. Friction Mechanism between GF/Rubber Composite and Metal

The forms of friction between GF/rubber composites and metals are mainly classified as abrasive and corrosive wear, and this study focuses on these two aspects:

(1) SiO_2_ particles have the characteristics of extremely easy agglomeration, while SiO_2_ particles are complex and irregular surfaces, so SiO_2_ particle aggregates are the leading cause of abrasive metal wear [38,39,40,41,42,43,44].

(2) The high-temperature ethanol–water mixed vapor will cause corrosion wear to the metal.

## 4. Experimental Results

### 4.1. Dispersion Analysis

#### 4.1.1. Payne Effect

The stress–strain curves of GF/rubber composites with different ratios are shown in Figure 7, and the Payne effects of GF/rubber composites with different ratios are shown in Table 5.

The data from Figure 7 were extracted in Table 5. From Table 5, it can be seen that the Payne effect of GF/rubber composites without TESPT addition was more significant. With the increase of GF content in the rubber matrix, the Payne effect of the composites first decreased and then increased. When the GF content was 3 phr, the Payne effect of the composite was the lowest. The Payne effect of the GF/rubber composite increased sharply when the added GF exceeded 3 phr.

GF has a unique spatial structure, and its surface can adsorb SiO_2_ particles in the rubber matrix. This promotes the dispersion of SiO_2_ particles in the rubber matrix and reduces the number of SiO_2_ particle agglomerates. Therefore, the Payne effect of GF/rubber composites decreases with GF addition. GF is a spatial filamentous structure, which is easy to agglomerate with each other in the rubber matrix, which hinders the dispersion of SiO_2_ particles. Meanwhile, the total surface area decreases after GF accumulation, and the adsorption of SiO_2_ particles is weakened. All of this leads to an ever-increasing Payne effect.

#### 4.1.2. Dispersion Degree of GF/Rubber Composite

From Figure 8 and Table 6, it can be seen that the GF/rubber composite had the lowest dispersion and the best dispersion when the content of GF was 3 phr. Adding too much or too little GF will increase the distribution of GF/rubber composites. This corresponds to the Payne effect of GF/rubber composites.

### 4.2. Silylation Reaction Index

The silylation reaction indices of GF/rubber composites with different ratios are shown in Figure 9. The silanization index is in Table 7.

It can be seen from Figure 9 that when the amount of GF added is 3 phr, the silanization index of the composite material is close to 1, and the silanization reaction is most advanced at this time. The silanization reaction of the composites with the remaining amount of GF was decreased. This is consistent with the previous analysis of the Payne effect.

### 4.3. Different Ratios of GF/Rubber Composite with Metal Friction and Wear

#### 4.3.1. Friction Coefficient

It can be seen from Figure 10 that the friction coefficient when the composite material rubs against the metal first decreased and then increased. The friction coefficient of GF/rubber composites was the smallest when the GF addition was 3 phr. It can be seen from Figure 10 that the friction coefficients of the experimental groups added with 0, 1, 5, and 7 phr GF increased by 22.4%, 18.5%, 20.8%, and 29.1%, respectively, relative to the experimental group with 3 phr of GF added. With the increase of GF content in the rubber matrix, SiO_2_ particles are continuously adsorbed on the surface of GF, which promotes the dispersion of SiO_2_ particles and reduces the number of SiO_2_ particle agglomerates in the rubber matrix. SiO_2_ particles have high physical hardness. The SiO_2_ particle aggregates have complex and irregular shapes, making the composite material have serious friction against the metal during the friction process, and the friction coefficient is significant. When the content of GF in the rubber matrix exceeded 3 phr, the aggregates of SiO_2_ particles in the rubber matrix continued to increase, which made the friction between the composite material and the metal intensify during the friction process and the friction coefficient increased.

#### 4.3.2. Metal Surface Topography

Figure 11 shows the height profile of the metal grinding head before and after sanding.

It can be seen from Figure 11 that a large number of contour peaks were ground after the metal surface of the C2 group was rubbed. There were many pits on the metal surface, and the wear was more serious. A part of the height profile peaks of the metal surface before and after friction in the C3 group was smoothed, which was reduced compared to the C2 group. After the metal surface of the C4 group was rubbed, a few height peaks were filed, the number of pits appeared less, and the wear was lighter. The metal surface of group C4 had more height peaks being ground after friction, the metal surface profile changed significantly, and the wear was more serious. The height profile of the metal surface before and after friction in the C6 group changed significantly, and a large number of height peaks were filed.

#### 4.3.3. Metal Wear

From Figure 12, it can be seen that with the increase of GF addition, the wear amount of GF/rubber composites on metal decreased and then increased, and the GF/rubber composite with a 3 phr GF addition had minor wear on metal. There was no adsorption of GF on SiO_2_ particles in the composite material without GF added, and the dispersion of SiO_2_ particles could not be promoted, so the metal wear was severe.

During the mixing process of the composites with 1 part of GF added, some SiO_2_ particles were adsorbed on the surface of GF, which reduced the number of SiO_2_ particle agglomerates in the composites. The SiO_2_ particle aggregates have complex and irregular shapes, making the composites produce severe friction with the metal during the friction process. With the reduction of SiO_2_ particle agglomerates in the composites, the abrasive wear of the composites to metals was gradually weakened. With the increase of the degree of the silanization reaction, the amount of high-temperature ethanol–water vapor increased, which increased the proportion of corrosion wear.

When 3 phr of GF was added, the surface area of GF that could adsorb SiO_2_ particles increased, and GF could adsorb more SiO_2_ particles, which better promoted the dispersion of SiO_2_ particles and further reduced the number of silica aggregates. Therefore, the GF/rubber composite with 1 phr of GF and the GF/rubber composite with 3 phr of GF had minor wear on the metal. As can be seen from Figure 12, the experimental group added 3 phr of GF, relatively. The wear amount of the experimental groups with 0, 1, 5, and 7 phr of GF added increased by 60%, 38.7%, 116%, and 146.7%, respectively.

#### 4.3.4. Wear Ratio

The abrasive wear amount of the metal grinding head is shown in Figure 13.

No high-temperature steam was sprayed during this experiment, and there was no corrosion and wear of metal, so the data measured in this experiment were the amount of abrasive wear. As can be seen from Figure 13, the experimental group added 3 phr of GF, relatively. The abrasive wear of the experimental groups with 0, 1, 5, and 7 phr of GF increased by 66.7%, 42%, 124.6%, and 165.2%, respectively. Compared with the above experiment, the wear ratio was obtained, and the wear ratio image is shown in Figure 14.

The increase in the degree of silanization reaction increased the amount of high-temperature ethanol–water mixed vapor, which increased the proportion of corrosive wear. When the GF addition amount was 3 phr, the contact area between GF and SiO_2_ particles reached the maximum, and the SiO_2_ particles were best dispersed. Therefore, the GF/rubber composite with a 3 phr GF addition had the most significant degree of silanization reaction and the highest percentage of corrosion and wear. Compared with the experimental groups with 0, 1, 5, and 7 phr of GF added, the corrosion wear rate with 3 phr of GF increased by 115.2%, 56.4%, 116.8%, and 748%, respectively.

#### 4.3.5. Roughness

From Figure 15, it can be seen that the amount of roughness change on the metal surface decreased and then increased with the increase of GF addition, which is related to the adsorption of GF to SiO_2_ particles. As can be seen from Figure 15, the experimental group added 3 phr of GF, relatively. The roughness variation of the experimental groups with 0, 1, 5, and 7 phr of GF increased by 191.8%, 97.4%, 103%, and 232%, respectively. With the increase of GF addition, the number of silica aggregates decreased. Silica is more complex and wears seriously on metal, so the surface roughness of metal gradually decreased. When the content of GF in the composite matrix exceeded 3 phr, the agglomeration of SiO_2_ particles in the composite material increased continuously, and the wear of the composite material to the metal increased. This resulted in severe wear on the metal surface and increasing roughness variation.

## 5. Conclusions

This study found that GF has a large surface area, and its surface can adsorb nano-SiO_2_ particles. With the increase of GF content in the composite matrix, the degree of the silanization reaction of the composite material first increased and then decreased. The amount of high-temperature ethanol–water mixed vapor corresponds to the degree of the silanization reaction. The proportion of abrasive wear of GF/rubber composites to metals decreased and then increased, and the proportion of corrosive wear increased and then decreased. With the increase of GF addition, the contact area between GF and SiO_2_ particles gradually increased, the dispersion of SiO_2_ particles was enhanced, the number of silica aggregates decreased, and the amount of GF/rubber composites to metal wear decreased. When the addition amount of GF was 3 phr, the GF/rubber composite had the lowest wear on metal. When the addition amount of GF exceeded 3 phr, there was a large amount of accumulation between GF, the contact area between GF and SiO_2_ particles gradually decreased, the dispersion of SiO_2_ particles decreased, the number of silica aggregates increased, and the wear amount of GF/rubber composites on metal increased. In the long run, the strengthening and transforming infrastructure in the Middle East and Asia-Pacific will increase the demand for GF. This study plays a vital role in reducing the wear of the internal mixer, improving the life of the internal mixer, and ensuring the performance of GF-modified rubber.

## Figures and Tables

**Figure 1 polymers-14-02849-f001:**
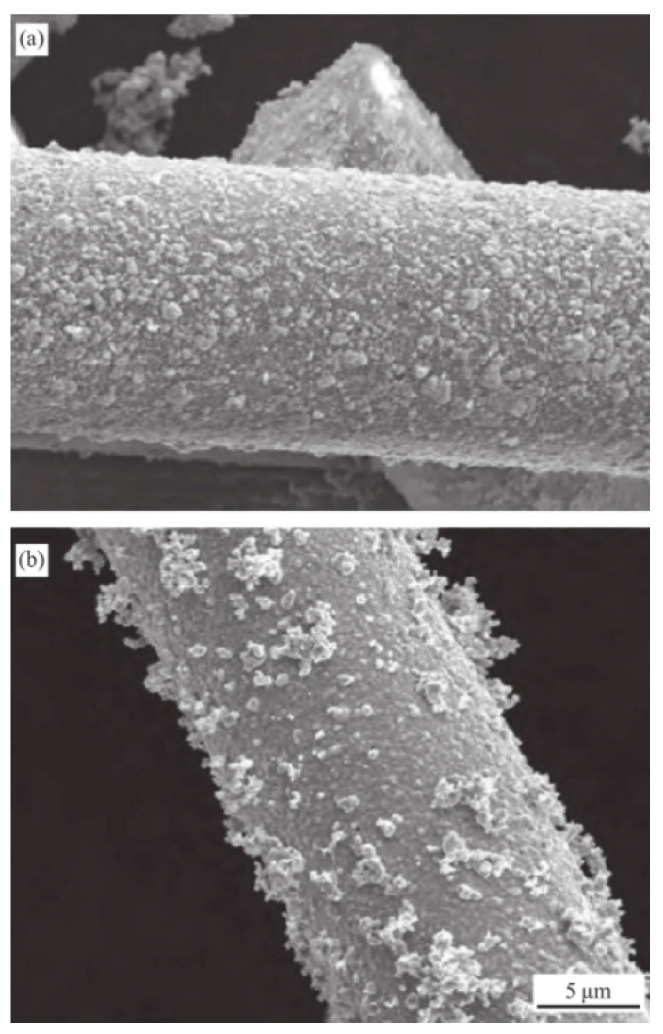
SEM image of GF. ((**a**) magnification is 10 μm, (**b**) magnification is 5 μm).

**Figure 2 polymers-14-02849-f002:**
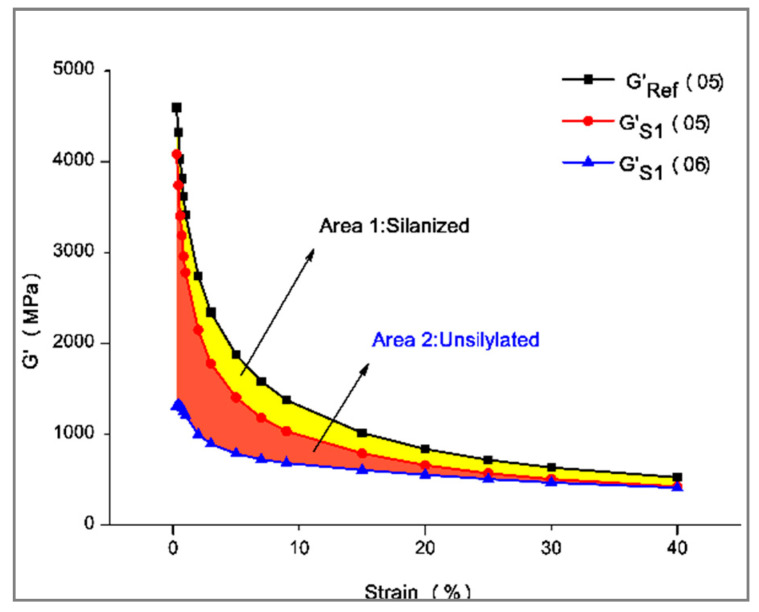
Distribution of silanization reaction degree.

**Figure 3 polymers-14-02849-f003:**
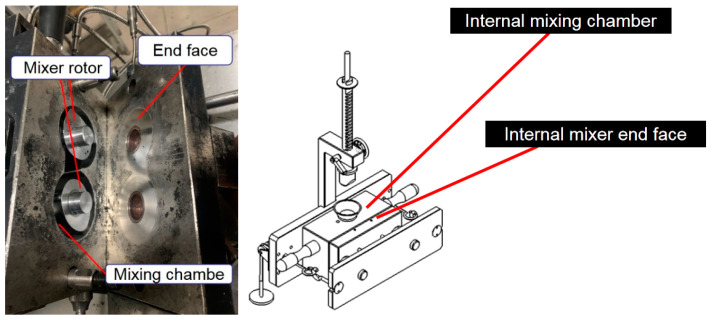
Hake internal mixer.

**Figure 4 polymers-14-02849-f004:**
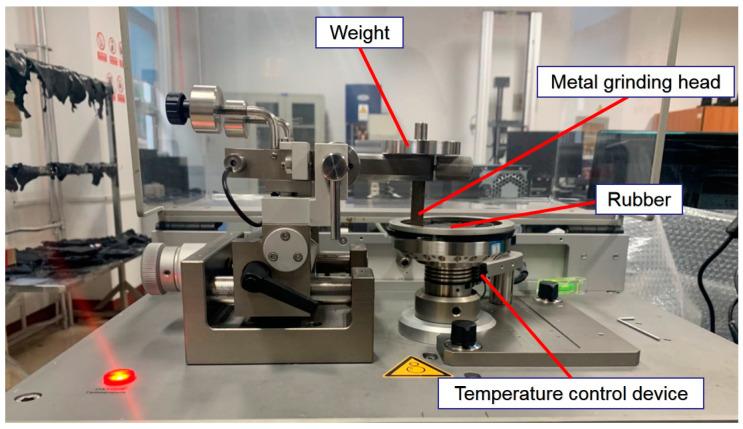
CSM friction and wear tester.

**Figure 5 polymers-14-02849-f005:**
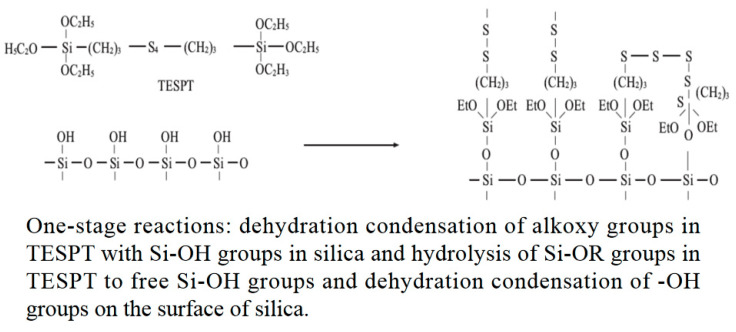
One-stage reactions.

**Figure 6 polymers-14-02849-f006:**
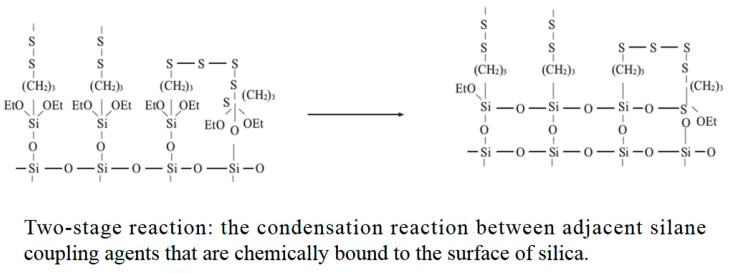
Two-stage reaction.

**Figure 7 polymers-14-02849-f007:**
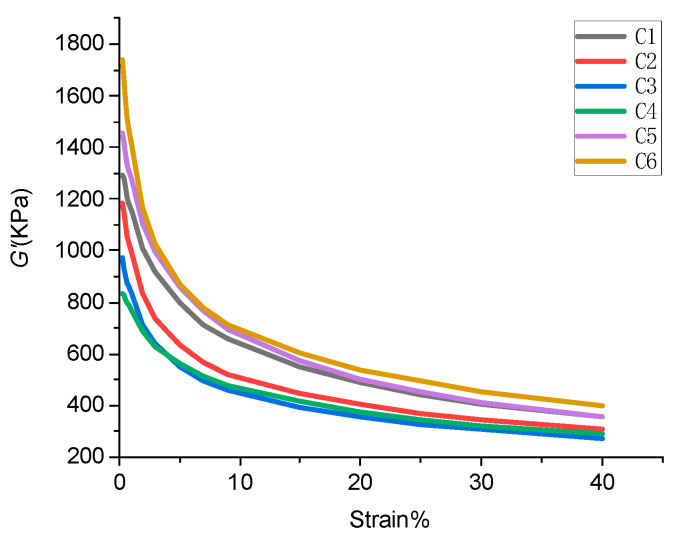
Stress–strain curves of GF/rubber composites with different ratios.

**Figure 8 polymers-14-02849-f008:**
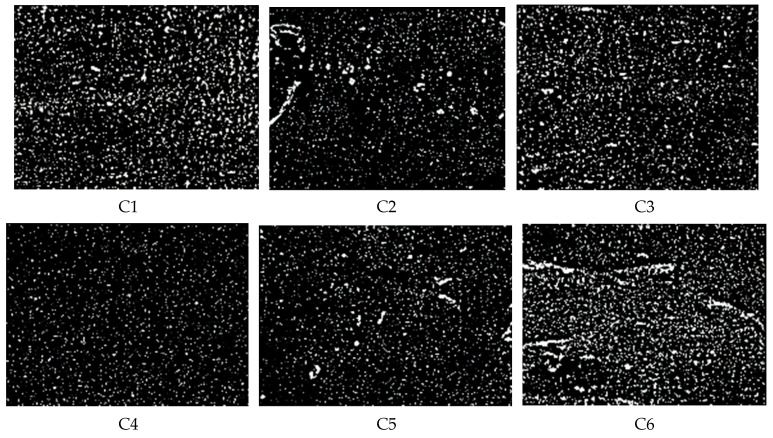
Dispersion images (the scale bar is 100:1).

**Figure 9 polymers-14-02849-f009:**
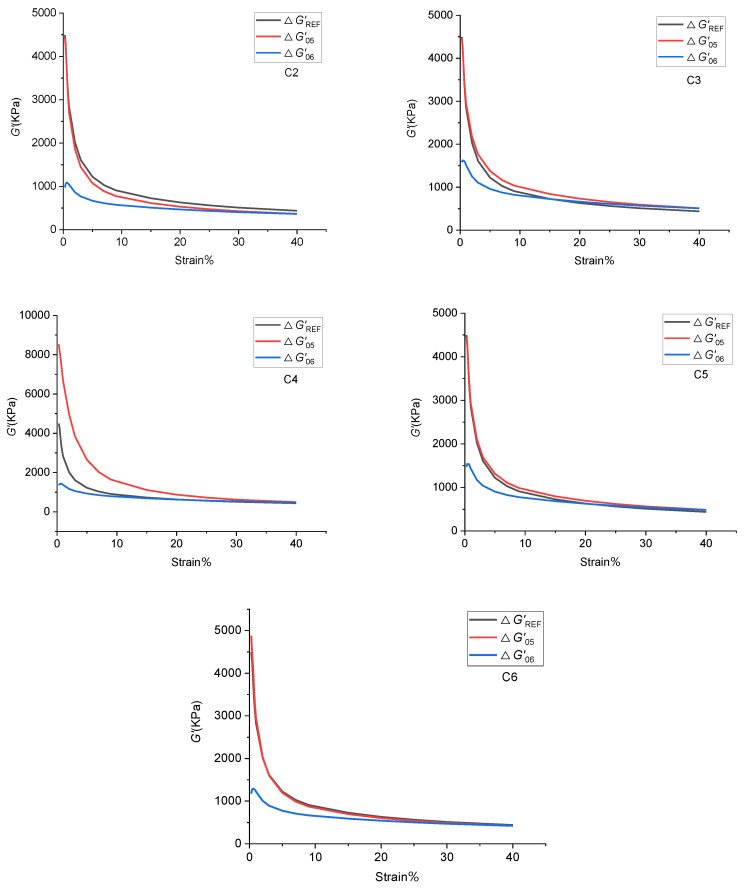
Silylation reaction image.

**Figure 10 polymers-14-02849-f010:**
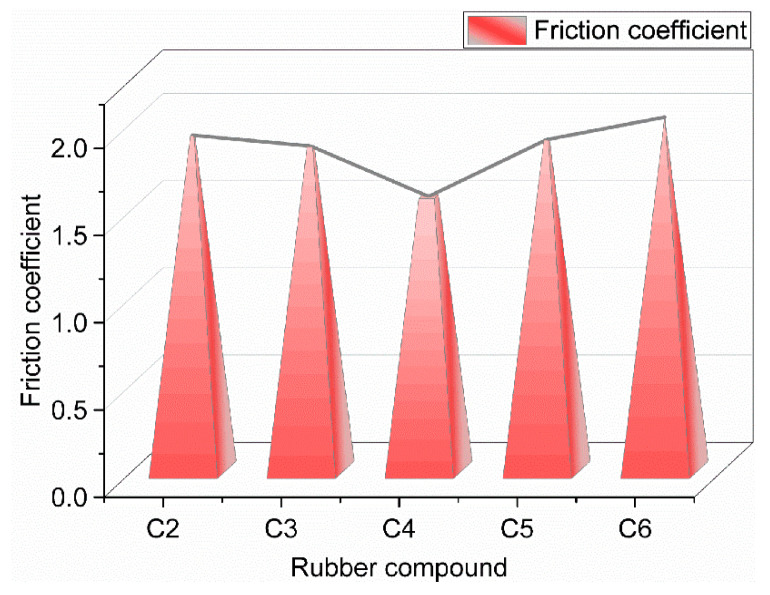
The average coefficient of friction of GF/rubber composites with different ratios against metal.

**Figure 11 polymers-14-02849-f011:**
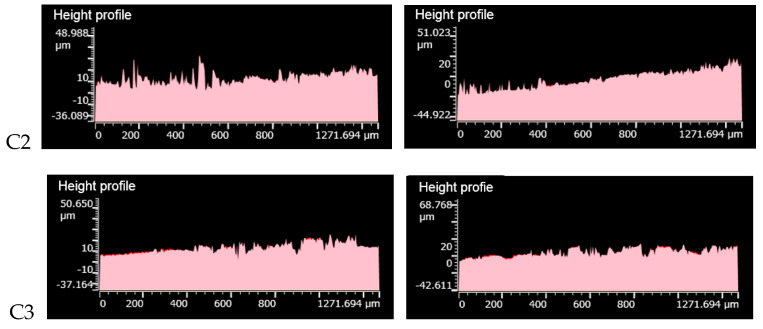
Height profile of metal grinding head before and after friction.

**Figure 12 polymers-14-02849-f012:**
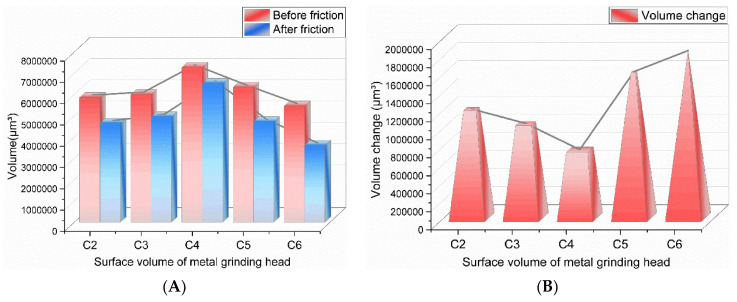
Metal wear volume. ((**A**) is the volume of the metal before and after friction, and (**B**) is the volume change).

**Figure 13 polymers-14-02849-f013:**
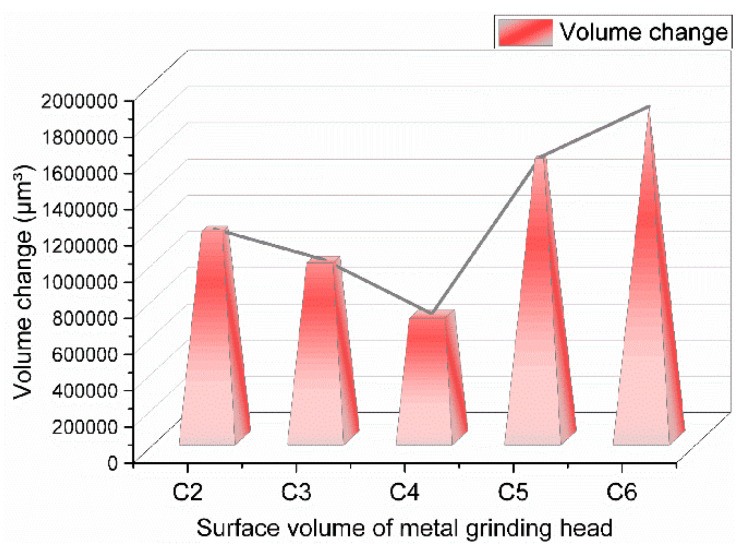
Metal wear volume without spraying high-temperature ethanol–water vapor mixture.

**Figure 14 polymers-14-02849-f014:**
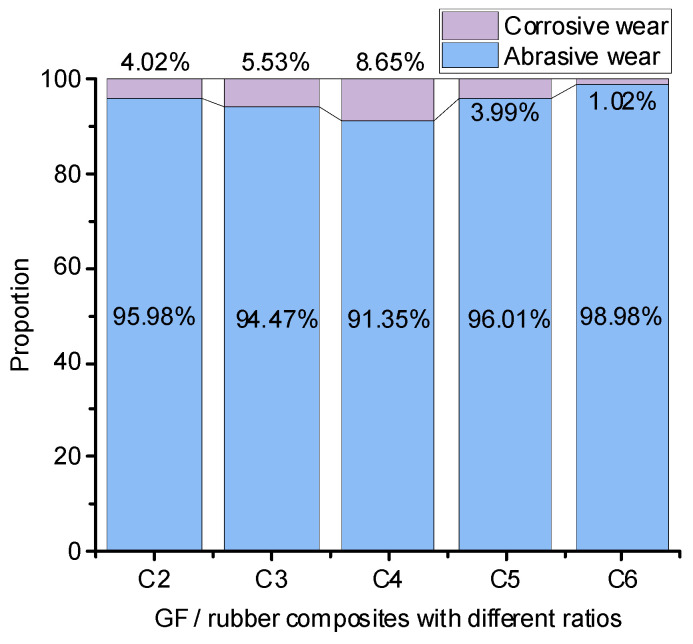
The proportion of abrasive wear and corrosion wear.

**Figure 15 polymers-14-02849-f015:**
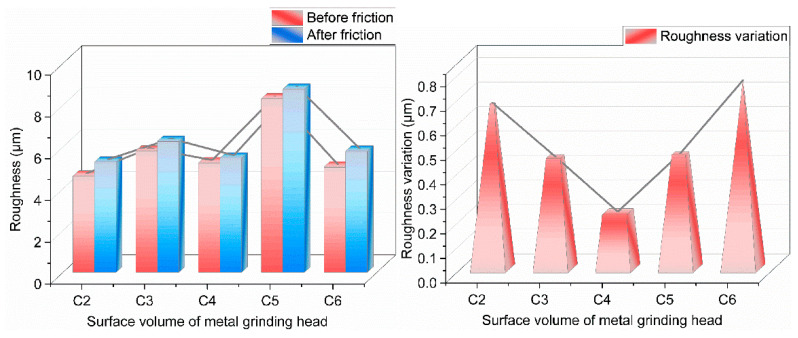
Metal surface roughness.

**Table 1 polymers-14-02849-t001:** Experimental instruments.

Experimental Instruments	Country of Production	Manufacturers
Hake mixer	China	Qingdao University of Science and Technology
BL-6157 double-roll rolling mill	China	Dongguan Baolun Precision Testing Instrument Co.
DisperGRADER Dispersion Meter	America	Alpha Corporation
LEXT OLS5000 3D laser measurement microscope	Japan	Products of Olympus Corporation
CSM-friction and abrasion tester	Switzerland	Tribometer Corporation
RPA2000 Rubber Processability Analyzer	America	Alpha Corporation
ZT-2588S Steam Generator	China	Zhiteng Company

**Table 2 polymers-14-02849-t002:** Formulation.

Raw Material	C1	C2	C3	C4	C5	C6
TSR20	15	15	15	15	15	15
BR9000	25.5	25.5	25.5	25.5	25.5	25.5
RC2557S	82	82	82	82	82	82
Silica115MP	45	45	45	45	45	45
GF	0	0	1	3	5	7
ZnO	2	2	2	2	2	2
TESPT	0	6	6	6	6	6
DPG	0.8	0.8	0.8	0.8	0.8	0.8
SAD	2	2	2	2	2	2
4020	2	2	2	2	2	2
CZ	1.8	1.8	1.8	1.8	1.8	1.8
S	1.3	1.3	1.3	1.3	1.3	1.3

**Table 3 polymers-14-02849-t003:** Mechanical blending process.

Hake Mixer		
Time (s)	T (°C)	Ingredients
0	70	BR9000, RC2557S, TSR20
40		TESPT, GF, DPG, SAD, 4020, ZnO, 1/2 Silica
70		1/2 Silica
150	120	Sweep
240	135	Sweep, Sampleing
300	145	Discharge

**Table 4 polymers-14-02849-t004:** Testing method.

Stage	Frequency/Hz	T/°C	Strain	Test Items
1	0.1	60	0.28%	* G’ * (01)
2	1	60	0.28–40%	* G’ * (02)
3	1	60	0.28–40%	* G’ * (03)
4	0.1	60	0.28%	* G’ * (04)
5	1	60	0.28–40%	* G’ * (05)
6	1	60	0.28–40%	* G’ * (06)

**Table 5 polymers-14-02849-t005:** Payne effect.

Formula	C1	C2	C3	C4	C5	C6
Payne effect	939.19	875.45	700.99	547.01	1099.81	1339.08

**Table 6 polymers-14-02849-t006:** Dispersion values.

Rubber Compounds	C1	C2	C3	C4	C5	C6
Dispersion	5.28	5.97	6.58	7.78	6.61	4.69

**Table 7 polymers-14-02849-t007:** Silylation reaction index.

Compounding Rubber	C2	C3	C4	C5	C6
Silylation reaction index	0.339	0.532	0.819	0.331	0.081

## Data Availability

The data presented in this study are available on request from the corresponding author.

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
