# Peer review of "Research on Friction and Wear Properties of Rubber Composites by Adding Glass Fiber during Mixing"

_polymers, 2022, doi:10.3390/polym14142849_

Round 1
Reviewer 1 Report
As I have already stated regarding the first version of the manuscript, your work assesses very well the behavior of the glassfiber-rubber composites and emphasizes the role played by the glass fibers in the friction and wear properties of the composites made with rubber, while the tables and explanations introduced in the second version of the article complete your work.
Author Response
Thank you for your support and encouragement to me. I sincerely wish you smooth work and happy life.
Reviewer 2 Report
Revised article is OK.
Author Response
Thank you for your support and encouragement to me. I sincerely wish you a smooth work and a happy life.

Reviewer 3 Report
The manuscript deals with the effect of glass fiber on the friction and wear properties of silica-filled rubber composites. The manuscript requires major overhauling before it gets accepted in polymers. Below are some specific points where major modifications are needed.
1) There are several spelling errors and grammatical and contextual errors in the paper. A thorough revision is needed.
2) The references mentioned in the “Introduction” section are both on plastics, such as PP and PI (page 2, L 42, L47). The authors should cite some work on rubber-GF composites. There are numerous research works done on GF-based composites available with silicone rubber, NBR, CR, EPDM, etc.
3) What was the reason for choosing a blend of BR, SBR, and NR at a particular ratio for the study? The authors should clarify that.
4) The “Experimental” section should have the following:
a. The names of the suppliers/producers are to be provided for the rubbers as the tradenames are used.
b. For silica: provide the details such as grade name, type (precipitated or fume), primary particle sizes, etc.
c. For GF: What was the average diameter and length? Was the GF surface-treated?
d. Why was the level of GF restricted up to 7 wt.%?
e. How was the rubber composite cured?
f. Is the measurement of “silylation reaction index” through RPA a standard test method?
5) Captions of Figures -5 & 6 and the associated texts are to be corrected. I assume, the authors indicated the “first stage” and “second stage” reactions.
6) Page 9, Table 6: What is the unit of dispersion? How was it calculated?
7) Page 9, Figure 8: Add a reference scale in the dispersion images. How representative are those images?
8) Page 11, Figure 11: Add a reference scale in the images. It is hard to differentiate the images before and after friction within the same set. I am unsure of the justification for providing them.
9) Page 11, L 263. The comment …” metal surface expanded significantly…” is unclear from the evidence images.
10) Figure 13: The scale of the images should be adjusted to provide more distinctive observations. The figures for C5 show more surface imperfections than C6. What is the reason?
11) Page 17, Figure 17. How was the roughness measured?
12) Overall, the “results and discussion” section is technically weak.
Author Response
Dear reviewer#3:
Thank you again for your careful and patient review. Your comments gave me a new understanding of the subject and new ideas. Thank you very much. My modifications are as follows.
Question (1): There are several spelling errors and grammatical and contextual errors in the paper. A thorough revision is needed.
Response to comments:
Thanks for your comments. I have polished the language of the article.
Question (2): The references mentioned in the “Introduction” section are both on plastics, such as PP and PI (page 2, L 42, L47). The authors should cite some work on rubber-GF composites. There are numerous research works done on GF-based composites available with silicone rubber, NBR, CR, EPDM, etc.
Response to comments:
Thanks for your opinion. I made changes to the "Introduction" section.
Li Hong [6] reviewed the development of continuous glass fibers in recent years, focusing on high-performance glass fibers. In particular, the article presents recent advances in the characterization of glass structures using NMR spectroscopy, Raman spectroscopy, and molecular dynamics simulation techniques.
Yang Chuanqi [7] used the mechanical blending method to prepare glass fiber (GF)/ natural rubber (NR)/butadiene rubber (BR) composites and studied the influence of glass fiber partially replacing silica on the vulcanization properties, mechanical properties, wear-resistance, and skid resistance of composites. The performance of the filler was compared with that of silica and nanometer attapulgite. The results show that adding glass fiber can shorten the curing time and increase the compound's hardness, tear strength, and wet friction coefficient. When the amount of glass fiber is 6 phr, the comprehensive performance of the binder is higher than that of silica and nanometer attapulgite filler.
Wang Wenjie [8] used glass fiber (GF) as filler and nitrile butadiene rubber (NBR) as a matrix to prepare NBR/GF composites by mechanical blending method and studied the effect of the amount of glass fiber on the properties of the composites. It was found that with the increase of glass fiber dosage, the vulcanization rate of the compound decreased gradually, and the tensile strength and 100% fixed elongation stress of the vulcanized compound increased first and then decreased. The maximum tensile strength and 100% fixed elongation stress were increased by 33% and 550%, respectively.
Chen Jiabin [9] prepared glass fiber/silicon rubber composite materials, studied the failure characteristics of glass fiber/silicon rubber composite materials by tensile test, analyzed the stress/strain curve of the composite material, and used the finite element method to simulate the composite material numerically. The study found that the glass fiber/silicone rubber composite specimen mainly manifested as fiber debonding fracture with fiber pulling out under the action of tensile load.
Lv Fuling [10] prepared nano-magnesium hydroxide/glass fiber/silicon rubber composites by mechanical blending. The dispersion state of nano-fillers in silicone rubber matrix was investigated, and the changing laws of physical and mechanical properties, heat-resistance, and flame-retardant properties of composites with different filler dosages were compared. The results show that additional magnesium hydroxide and glass fiber dosage ratios can improve silicone rubber's physical and mechanical properties. Among them, tensile strength, tear strength, compression set, and hardness have been enhanced. Combining the two fillers improves the composite's thermal stability and flame retardant properties to a certain extent.
Hao Qiangqiang [11] prepared glass fiber/nitrile rubber composites by mechanical blending method, studied the effect of glass fiber content and length on the wear resistance and mechanical properties of the composites and observed the wear surface of the composites with a scanning electron microscope. The results show that the optimum addition mass of chopped glass fibers in the nitrile rubber matrix is 3phr, and the length is 6mm. Under these conditions, the composites prepared have the best wear resistance and comprehensive mechanical properties, and the wear surface is the flattest and smooth.
Question (3): What was the reason for choosing a blend of BR, SBR, and NR at a particular ratio for the study? The authors should clarify that.
Response to comments:
Thank you for your patience and careful review. I will explain it to you in detail.
Butadiene Rubber 9000 (BR9000), China Hainan Natural Rubber Industry Group Co., Ltd.; Polymeric Styrene Butadiene Rubber (RC2557S), Dongguan Futai Rubber Trading Co., Ltd., China; Natural Rubber (TSR20), China Hainan Natural Rubber Industry Group Co., Ltd. Company; Bis-[γ-(triethoxysilyl)propyl]tetrasulfide (TESPT), Antioxidant 4020, Zinc Oxide ZnO, Stearic Acid SAD, 1,3-Diphenylguanidine (DPG) , accelerator CZ, and sulfur S are products of China Henan Longji Chemical Co., Ltd.; Glass Fiber (GF), Wanlong Composite Materials Co., Ltd; Silica115MP has a specific surface area of 115 m2/g, 7-40 nm particle size, and solid powder, Solvay Silica Co., Ltd; GF diameter is 12-23 μm, length is 0.2 mm-0.6 mm, and the surface is not treated.
Question (4): The “Experimental” section should have the following:
- The names of the suppliers/producers are to be provided for the rubbers as the tradenames are used.
- For silica: provide the details such as grade name, type (precipitated or fume), primary particle sizes, etc.
- For GF: What was the average diameter and length? Was the GF surface-treated?
- Why was the level of GF restricted up to 7 wt.%?
- How was the rubber composite cured?
- Is the measurement of “silylation reaction index” through RPA a standard test method?
Response to comments:
Thanks for your comments. I will answer your questions one by one.
Butadiene Rubber 9000 (BR9000), China Hainan Natural Rubber Industry Group Co., Ltd.; Polymeric Styrene Butadiene Rubber (RC2557S), Dongguan Futai Rubber Trading Co., Ltd., China; Natural Rubber (TSR20), China Hainan Natural Rubber Industry Group Co., Ltd. Company; Bis-[γ-(triethoxysilyl)propyl]tetrasulfide (TESPT), Antioxidant 4020, Zinc Oxide ZnO, Stearic Acid SAD, 1,3-Diphenylguanidine (DPG) , accelerator CZ, and sulfur S are products of China Henan Longji Chemical Co., Ltd.; Glass Fiber (GF), Wanlong Composite Materials Co., Ltd; Silica115MP has a specific surface area of 115 m2/g, 7-40 nm particle size, and solid powder, Solvay Silica Co., Ltd. GF diameter is 12-23 μm, length is 0.2 mm-0.6 mm, and the surface is not treated.
b.Silica115MP has a specific surface area of 115 m2/g, 7-40 nm particle size, and solid powder, Solvay Silica Co., Ltd.
c.GF diameter is 12-23 μm, length is 0.2 mm-0.6 mm, and the surface is not treated.
- Because when more GF is added, the accumulation of GF in the rubber matrix is severe, and GF particles' dispersion is poor, reducing the rubber properties.
e.The width of the twin rolls was set to 8 mm, and the GF/rubber composite material kneaded by the internal mixer was placed between the two rolls of the twin roll press. After many times of rolling, a cured rubber with a flat surface thickness of 8 mm was obtained. An image of the twin roll press is shown in Figure 1.
Figure 1. Twin Roll Press
- The silanization reaction index test is a commonly used test method in the rubber field and is a standardized test method.
Question (5): Captions of Figures -5 & 6 and the associated texts are to be corrected. I assume, the authors indicated the “first stage” and “second stage” reactions.
Response to comments:
Yes, it has been pointed out in the text. Thanks for your careful review.
Question (6): Page 9, Table 6: What is the unit of dispersion? How was it calculated?
Response to comments:
Thank you for your comments. I will explain to you in detail the testing principle of dispersion. The evaluation method of the degree of distribution is to obtain the degree of dispersion by comparing it with a standard image, so the degree of dispersion has no unit.
Products of Alpha Company in the United States. The outline diagram and detection process of Alpha's carbon black dispersion detection equipment is shown in Figure 2. First, the cutting surface of the rubber is obtained with a cutter, and then the image is received with an optical system. The software analyzes and calculates the painting according to the standard ISO 11345 and ASTM D7723, and finally obtains the dispersion level of the packing.
Figure 2.Filler dispersion tester
The evaluation method of the dispersion is to get the dispersion level by comparing it with the standard image. The specific process is as follows: Cut the rubber open to expose a fresh surface. Under a light source with an inclination Angle of 30°, the section is observed with an optical or photographic microscope to obtain a sample surface image magnified 100 times. Then, this image is compared with the standard reference image of 10 grades to get the dispersibility grade X of the sample. Level 10 means the best dispersion, and the model is close to optimal physical properties, while level 1 indicates the distribution is very poor. Generally, there is a relationship between the dispersion level and the properties of the compound, as shown in Table 1. A standard picture of the 10 levels is shown in Figure 3.
Table 1. The relationship between dispersion grade and the properties of rubber
Dispersion grade |
Performance classification |
9-10 |
Excellent |
8 |
Good |
7 |
Acceptable |
5-6 |
Unqualified |
3-4 |
Difference |
1-2 |
Very bad |
Figure 3. Standard pictures of dispersion levels 1-10
Question (7): Page 9, Figure 8: Add a reference scale in the dispersion images. How representative are those images?
Response to comments:
Thanks for your input. I have added it. The dispersion degree image is a random test, five images are obtained in one test, and the system automatically takes the average value, so it has good representativeness.
C1 |
C2 |
C3 |
C4 |
C5 |
C6 |
Figure 8. Dispersion images (The scale bar is 100:1)
Question (8): Page 11, Figure 11: Add a reference scale in the images. It is hard to differentiate the images before and after friction within the same set. I am unsure of the justification for providing them.
Response to comments:
Thanks for your valuable comments. After careful consideration, I removed Figure 11.
Question (9): Page 11, L 263. The comment …” metal surface expanded significantly…” is unclear from the evidence images.
Response to comments:
Thanks for your comments. I have removed the content related to Figure 11.
Question (10): Figure 13: The scale of the images should be adjusted to provide more distinctive observations. The figures for C5 show more surface imperfections than C6. What is the reason?
Response to comments:
Thanks for your opinion. But since the device exports the image, I can't change the scale. Therefore I chose to delete Figure 13.
Question (11): Page 17, Figure 17. How was the roughness measured?
Response to comments:
The LEXT OLS5000 3D laser measuring microscope can scan the small spacing and tiny peaks and valleys of the metal surface by laser and directly obtain the metal surface roughness through analysis. Wipe the metal surface with alcohol cotton before rubbing, and observe the roughness of the metal before rubbing with a LEXT OLS5000 3D laser measuring microscope. After rubbing, wipe the metal surface with alcohol cotton, and continue to measure the roughness after rubbing.
For your convenience, I took a picture of the device.
Overall equipment Overall equipment
Question (12): Overall, the “results and discussion” section is technically weak.
Response to comments:
Thanks for your opinion. I modified this part. I've added the relevant data and blued it.
It can be seen from Figure 10 that the friction coefficient when the composite material rubs against the metal first decreases and then increases. The friction coefficient of GF/rubber composites is the smallest when the GF addition is 3 phr. It can be seen from Figure 10 that the friction coefficients of the experimental groups added with 0, 1, 5, and 7 phr GF increased by 22.4%, 18.5%, 20.8%, and 29.1%, respectively, relative to the experimental group added with 3 phr GF. With the increase of GF content in the rubber matrix, SiO2 particles are continuously adsorbed on the surface of GF, which promotes the dispersion of SiO2 particles and reduces the number of SiO2 particle agglomerates in the rubber matrix. SiO2 particles have highly high physical hardness. The SiO2 particle aggregates have complex and irregular shapes, making the composite material have serious friction against the metal during the friction process, and the friction coefficient is significant. When the content of GF in the rubber matrix exceeds 3 phr, the aggregates of SiO2 particles in the rubber matrix continue to increase, which makes the friction between the composite material and the metal intensify during the friction process and the friction coefficient increases.
From Figure 12, it can be seen that with the increase of GF addition, the wear amount of GF/rubber composites on metal decreases and then increases, and the GF/rubber composite with 3 phr GF addition has minor wear on metal. There is no adsorption of GF on SiO2 particles in the composite material without GF added, and the dispersion of SiO2 particles cannot be promoted, so the metal wear is severe.
During the mixing process of the composites added with 1 part of GF, some SiO2 particles were adsorbed on the surface of GF, which reduced the number of SiO2 particle agglomerates in the composites. The SiO2 particle aggregates have complex and irregular shapes, making the composites produce severe friction with the metal during the friction process. With the reduction of SiO2 particle agglomerates in the composites, the abrasive wear of the composites to metals is gradually weakened. With the increase of the degree of silanization reaction, the amount of high-temperature ethanol-water vapor increases, which increases the proportion of corrosion wear.
When 3 phr GF was added, the surface area of GF that could adsorb SiO2 particles increased, and GF could adsorb more SiO2 particles, which better promoted the dispersion of SiO2 particles and further reduced the number of silica aggregates. Therefore, the GF/rubber composite with 1 phr GF and the GF/rubber composite with 3 phr GF has minor wear on the metal. As can be seen from Figure 12, the experimental group added 3 phr GF relatively. The wear amount of the experimental groups added with 0, 1, 5, and 7 phr GF increased by 60%, 38.7%, 116%, and 146.7%, respectively.
No high-temperature steam is sprayed during this experiment, and there is no corrosion and wear of metal, so the data measured in this experiment is the amount of abrasive wear. As can be seen from Figure 13, the experimental group added 3 phr GF relatively. The abrasive wear of the experimental groups with 0, 1, 5, and 7 phr GF increased by 66.7%, 42%, 124.6%, and 165.2%, respectively. Compared with the above experiment, the wear ratio is obtained, and the wear ratio image is shown in Figure 14.
The increase in the degree of silanization reaction increases the amount of high-temperature ethanol-water mixed vapor, which increases the proportion of corrosive wear. When the GF addition amount is 3 phr, the contact area between GF and SiO2 particles reaches the maximum, and the SiO2 particles are best dispersed. Therefore, the GF/rubber composite with 3 phr GF addition has the most significant degree of silanization reaction and the highest percentage of corrosion and wear. Compared with the experimental groups with 0, 1, 5, and 7 phr GF added, the corrosion wear rate with 3 phr GF increased by 115.2%, 56.4%, 116.8%, and 748%, respectively.
From Figure 15, it can be seen that the amount of roughness change on the metal surface decreases and then increases with the increase of GF addition, which is related to the adsorption of GF to SiO2 particles. As can be seen from Figure 15, the experimental group added 3 phr GF relatively. The roughness variation of the experimental groups with 0, 1, 5, and 7 phr GF increased by 191.8%, 97.4%, 103%, and 232%, respectively. With the increase of GF addition, the number of silica aggregates decreased. Silica is more complex and wears seriously on metal, so the surface roughness of metal gradually decreases. When the content of GF in the composite matrix exceeds 3 phr, the agglomeration of SiO2 particles in the composite material increases continuously, and the wear of the composite material to the metal increases. This results in severe wear on the metal surface and increasing roughness variation.

This manuscript is a resubmission of an earlier submission. The following is a list of the peer review reports and author responses from that submission.
Round 1
Reviewer 1 Report
Your paper proves once more the importance of pretreatment on the surfaces of two phases that are put in contact to work together. As literature suggests, and your study confirms, GF monofilaments are very easy to agglomerate each other, but they have a large surface area, and this can adsorb numerous nano-SiO2 particles, which can promote their dispersion and reduce the accumulation of SiO2 particles, while the degree of silanization reaction of GF/rubber composites increases, thus decreasing the wear on metal.
Reviewer 2 Report
This work is not well presented in terms of clarity and consistence. The experimental details are not provided clearly, lots of initials are confusing. The results and discussions are not well-organised, which makes it hard to follow the results.
- Figure 1 is unnecessary, it does not provide any dimension nor surface properties of the GF, a digital image is not helpful
- is the HAKE another version of HAAKE? In Table 1 the 'Manufacurer' column, the words' products' should be removed.
- the initials in Table 2 should be explained, full names should be provided.
- the information in Table 3 is rather confusing, what are the polymers, chemicals, what the unit of the time......?
- Figure 5 is unnecessary, as it appears/repeats in Figure 6 as well.
- the images in Figure 11-13 are doubtable, without consistent comparsion conditions.
Reviewer 3 Report
There are many points to be revised and reconsidered because of many lack of explanation and of many deviation from journal style.
